# Integrated Management of Bacterial Wilt and Root-Knot Nematode Diseases in Pepper: Discovery of Phenazine-1-Carboxamide from *Pseudomonas aeruginosa* W-126

**DOI:** 10.3390/ijms26073335

**Published:** 2025-04-03

**Authors:** Shuai Wang, Yifan Wang, Youzhi Yao, Wenzhuo Li, Zhan Hu, Dong Li, Ranfeng Sun

**Affiliations:** Key Laboratory of Green Prevention and Control of Tropical Plant Diseases and Pests, Ministry of Education, School of Tropical Agriculture and Forestry, Hainan University, Haikou 570228, China; 996018@hainanu.edu.cn (S.W.);

**Keywords:** phenazine-1-carboxamide, *Pseudomonas aeruginosa*, *Ralstonia solanacearum*, *Meloidogyne incognita*, structure−activity relationship

## Abstract

*Ralstonia solanacearum* is an important pathogen causing bacterial wilt in pepper (*Capsicum annuum* L.). The concurrent infection of *R. solanacearum* and root-knot nematodes *(Meloidogyne* spp.) exacerbates the severity of bacterial wilt in pepper. Utilizing plant endophytic bacteria to control these mixed diseases is a viable strategy. *Waltheria indica* L. (Sterculiaceae) is a traditional medicine plant. A total of 209 endophytic bacteria were isolated from *W. indica*, and *Pseudomonas aeruginosa* W-126 showed an efficient antagonistic effect against *R. solanacearum*. Based on active compound tracking principles, a compound was isolated through silica gel column chromatography and preparative HPLC combined with TLC analysis. It was identified as phenazine-1-carboxamide (PCN) by spectral techniques (ESI-MS, ^1^H-NMR, ^13^C-NMR). PCN displayed excellent inhibitory activity against *R. solanacearum*, with an EC_50_ of 64.16 μg/mL in vitro. In addition, it showed certain nematocide activity, with an LC_50_ value of 118.63 μg/mL at 72 h. PCN also showed certain inhibitory activity against five other phytopathogenic bacteria. The structure−activity relationship indicated that the phenazine skeleton and acylamide groups were the key pharmacophores for the activity of phenazine-related compounds against *R. solanacearum*. PCN controlled the complex diseases of *R. solanacearum* and *M. incognita* in a pot experiment, with respective 51.41 and 39.80% inhibitory rates. The exploration of secondary metabolites of biocontrol bacteria can provide reference for the development of novel and efficient pesticides.

## 1. Introduction

*Ralstonia solanacearum* exhibits a widespread geographical distribution and remarkable genetic diversity, and it can affect over 450 plant species [1,2]. Bacterial wilt, the devastating soil-borne disease caused by *R. solanacearum*, is primarily found in tropical and subtropical regions. It can lead to significant economic losses, severely reducing the yield and quality of vegetable crops [3,4]. Root-knot nematodes, with a broad host range, are globally widespread and are responsible for approximately USD 100 billion in annual economic losses worldwide [5,6]. Infection by root-knot nematodes can exacerbate the occurrence of plant fungal and bacterial diseases. Bacterial wilt and root-knot nematode populations often coexist, and where root-knot nematodes are prevalent, so too is bacterial wilt [7,8]. The severity of bacterial wilt is directly related to the density of root-knot nematodes. Root-knot nematode infection creates entry points for *R. solanacearum* into plant roots [9,10]. The root knots caused by nematode infection provide ideal environments for bacterial growth, allowing large numbers of bacteria to colonize these areas. Therefore, controlling both bacterial wilt and root-knot nematodes is crucial in reducing the incidence and impact of this complex disease [10,11].

The use of biocontrol microorganisms to manage complex diseases such as bacterial wilt and root-knot nematodes is a feasible approach [12,13]. These microorganisms can produce active compounds that are toxic to both pathogens [14,15]. Screening active substances from natural products is one of the effective ways to develop new pesticides. However, the probability of obtaining abundant compounds from medicinal plants is limited due to resource constraints. Endophytes are currently being developed as important resources without this problem [16]. Due to the special condition of parasitism inside plants, endophytes can produce similar active compounds to host plants, and have great potential for development. Moreover, endophytes, as microbial resources, have the advantage of culturability, and target products can quickly be obtained from them through systematic separation and purification methods. This also facilitates industrial production possibilities. Consequently, the search for new active products and medicinal substances from the metabolites of endophytes not only overcomes the problems of the slow growth cycle and low yield of medicinal plants, but also significantly enriches pesticide resources, rendering this an area of extremely high research value and significance [17,18]. *Waltheria indica* L. is a traditional officinal plant with high medicinal values [19,20]. In addition, it produces many compounds against plant pathogens and root-knot nematodes [21]. However, whether its endophytes can also exhibit activity against plant pathogens and nematodes and whether they produce highly effective compounds has not been reported.

Several studies have reported that *Pseudomonas aeruginosa* can be used as a biocontrol agent to manage bacterial wilt [22,23,24] and root-knot nematodes [12,25,26]. A bacterial competition assay showed that HSI-I-type T6SS of *P. aeruginosa* VIH2 exhibited dramatic antibacterial activity against *R. solanacearum* [27]. The literature reports that N-acyl homoserine lactone (AHL) was degraded by the cell-free lysate of *P. aeruginosa* 2apa, and also inhibited the biofilm formation of *R. solanacearum* [28]. In addition to these degrading enzymes, it was unclear whether *P. aeruginosa* can produce other small active compounds to inhibit *R. solanacearum*. *P. aeruginosa* produces phenazine-1-carboxylic acid (PCA) and phenazine-1-carboxamide (PCN), which could effectively inhibit *R. solani* and *X. oryzae* pv. oryzae [29,30,31]. It has been found that PCN functionalizes mesoporous silica nanoparticles as antimicrobial coatings against *Candida albicans* [32]. These reports indicate that PCN possesses bactericidal potential. Nevertheless, the anti-bacterial wilt capability of PCN, as well as other phenazine compounds, has been infrequently reported. We screened out bacterium W-126 with an efficient antagonistic effect against *R. solanacearum* from 209 endophytic bacteria of *W. indica*. PCN was isolated based on active compound tracking principles from dozens of components of the bacterium W-126. Our study assesses the efficacy of PCN in managing the complex diseases caused by *R. solanacearum* and *M. incognita*, offering technical insights that support the use of PCN for controlling soil-borne diseases in pepper.

## 2. Results

### 2.1. Antibacterial Results of Endophytic Bacteria Against R. solanacearum

Using *R. solanacearum* as indicator bacteria, the antibacterial effect of 209 biocontrol bacteria was measured using the filter paper diffusion method. The inhibition zone of strain W-126 against *R. solanacearum* was the largest, with an average diameter of 26.7 mm (Figure 1, Appendix A).

### 2.2. Identification of Strain W-126

The growth of bacterium W-126 on the LB solid plate was normal, and morphological observation showed that its colony was light yellow, with a moist surface and slight uplift, and its texture was viscous and easy to pick up (Figure 2A). After Gram staining, all the bacteria were short, rod-shaped, and red (Figure 2B), indicating a Gram-negative bacterium.

The physiological and biochemical test results revealed that strain W-126 was oxidase-positive; it was capable of producing pyocyanin, liquefying gelatin, and reducing nitrate, and could grow at 42 °C (Appendix A). The sequence of strain W-126 was compared with the NCBI database, and similar sequences were identified using the BLAST tool (Version 1.4.0). The results showed that strain W-126 had very high homology with *P. aeruginosa* after phylogenetic tree analysis with MEGA 11 software (Figure 3). Based on its morphological, physiological, and biochemical characteristics, and on the phylogenetic tree analysis, the strain W-126 was identified as *P. aeruginosa*.

### 2.3. Isolation and Extraction of Bacteriostatic Substances from Strain W-126

The antibacterial activity of W-126 extracts from various organic solvents was determined. When the extract concentration was 0.3 mg/mL, inhibition zones appeared in all of the extracts (Figure 4), implying that *R. solanacearum* was inhibited at different degrees by the W-126 extracts. Notably, the ethyl acetate extract displayed the most prominent inhibitory zone, with a diameter of 10 mm (Figure 4B), indicating that ethyl acetate more effectively extracted the active compounds against *R. solanacearum*.

### 2.4. Inhibitory Effect on R. solanacearum and Identification of Compound W-s1

Ethyl acetate fractional extract was subsequently used as the separation fraction. Bioassay-guided column chromatography with instrumental analysis led to the isolation of the antibacterial effect compound (W-s1) from the ethyl acetate extraction. When the concentration of W-s1 was 1 mg/mL, the inhibition zone reached 25 mm, and there was an obvious inhibition zone with obvious inhibitory effect (Figure 5A).

W-s1: Yellow solid, identified as phenazine-1-carboxamide (PCN, Figure 5), m.p. 201~203 °C; 1H NMR (400 MHz, Chloroform-d) δ: 9.02 (dd, J = 7.1, 1.5 Hz, 1H), 8.46 (dd, J = 8.6, 1.5 Hz, 1H), 8.36–8.29 (m, 1H), 8.29–8.20 (m, 1H), 8.03–7.88 (m, 3H); 13C NMR (101 MHz, Chloroform-d) δ:166.66, 143.41, 143.08, 141.57, 140.85, 136.06, 134.35, 131.86, 131.22, 130.00, 129.71, 129.14, 128.76. ESI-MS: *m*/*z* 224.08 [M+H]^+^ (calcd C_13_H_9_N_3_O for 223.07).

### 2.5. Antibacterial Activity of PCN Against Phytopathogenic Bacteria and M. incognita

The activities of PCN against six phytopathogenic bacteria and *M. incognita* are shown in Table 1 and Figure 6. These show that PCN had a high inhibitory activity against *R. solanacearum*, with an EC_50_ of 64.16 μg/mL (Table 1). PCN also exhibited inhibitory activity against *R. solanacearum* when the concentration of PCN was 7.81 μg/mL (Figure 6). In addition, PCN showed excellent inhibitory activity against *X. campestris*, with an EC_50_ of 138.70 μg/mL, and certain nematicide activity, with an LC_50_ of 118.63 μg/mL at 72 h (Table 1).

### 2.6. Results of Activity of PCN Analogues Against R. solanacearum and M. incognita

All the treatment concentrations of PCN analogues were 125 μg/mL, and only the inhibitory rates of 3-amino-phenazin-2-ol, PCN, phenazine, phenazine methosulfate, and neutral red against *R. solanacearum* were over 60%. The inhibitory rate of phenazine against *R. solanacearum* was 66.33% (Table 2, Appendix A), which indicates that phenazine has excellent inhibitory activity against *R. solanacearum*. Although PCN was not the most active of these compounds, PCN showed the second highest inhibitory rate of all the PCN analogues against *R. solanacearum* (Table 2). For *M. incognita*, only the inhibitory rates of phenazin-1-ylamine, PCN, phenazine, and 1-Hydroxyphenazine were over 50% (Table 2).

### 2.7. Pot Experiment Results of PCN for Controlling Complex Diseases of R. solanacearum and M. incognita

The experiment was divided into seven treatment groups: pepper with neither *R. solanacearum* nor *M. incognita* (CK-), pepper with both *R. solanacearum* and *M. incognita* (CK), pepper with both *R. solanacearum* and *M. incognita* under PCN control (PCN), pepper with both *R. solanacearum* and *M. incognita* under the treatment of W-126 fermentation broth (W-126), pepper with both *R. solanacearum* and *M. incognita* under Zhuorun treatment (Zhuorun, the commercial biocontrol agent of *B. amyloliquefaciens* QST713 in China) as a positive control against the complex diseases of *R. solanacearum* and *M. incognita*, pepper with both *R. solanacearum* and *M. incognita* under abamectin treatment (Abamectin) as a positive control against *M. incognita,* pepper with both *R. solanacearum* and *M. incognita* under streptomycin sulfate treatment (streptomycin sulfate) as a positive control against *R. solanacearum*. The plant heights of all seven treatment groups showed no significant differences. The plant weight of CK- was significantly higher than the other six treatments (Table 3), which indicates that the complex diseases of *R. solanacearum* and *M. incognita* affected the weight of the peppers. The root-knot number of CK was significantly higher than the other four treatments of PCN, Zhuorun, W-126, and abamectin (Table 3, Appendix A), which indicates that PCN and Zhuorun could inhibit *M. incognita*. The control efficiencies of PCN, W-126, and Zhuorun against *M. incognita* were, respectively, 39.80, 54.31, and 60.16%, but they were all lower than the control efficiency of abamectin, which was 65.99% (Table 3). The bacterial wilt disease index of CK was significantly higher than the other six treatments, and the control efficiencies of PCN and streptomycin sulfate were, respectively, 51.41 and 57.76%, which were significantly higher than the treatments of W-126 and Zhuorun, with 37.57 and 30.58% (Table 3, Appendix A). These results indicate that PCN shows similar control efficiency against *R. solanacearum* with streptomycin sulfate when the complex diseases of *R. solanacearum* and *M. incognita* occur. In addition, the control efficiencies of these two compounds were higher than the biocontrol agent of Zhuorun.

## 3. Discussion

The concomitant infection of *R. solanacearum* and root-knot nematodes (*Meloidogyne* spp.) increases the severity of bacterial wilt in horticultural plants [7,8,10]. The use of biocontrol microorganisms to control these complex diseases is a feasible method [13,14,15]. The bacterial strains *Bacillus subtilis* AR12 and *B. subtilis* SM21 can control root-knot and bacterial wilt mixed diseases in bell pepper [33]. Manganese oxide nanoparticles (MnO_2_ NPs) and *Pseudomonas putida* alone or in combination have been found to reduce the complex diseases of *R. solanacearum* and *M. incognita* [13]. The n-Butanol extract of *Bacillus gottheilii* MSB1 has exhibited positive impacts against *R. solanacearum*, and ethyl acetate and n-Butanol extracts have demonstrated in vitro activity against *M. incognita* [15]. However, efficient active compounds against both *R. solanacearum* and *M. incognita* from one biocontrol microorganism have rarely been explored and reported.

*Waltheria indica* L. is a traditional medicine plant with high medicinal value [19,20]. In addition, it produces many compounds against plant pathogens and nematodes [21,34,35,36]. However, whether its endophytes can also exhibit activity against plant pathogens and nematodes and whether they produce highly effective compounds has not been reported. Due to the special condition of parasitism inside plants, endophytes can produce similar active compounds to host plants [16]. In addition, endophytes have the characteristic of culturability, and target products can quickly be obtained from them. Consequently, the search for new active products and medicinal substances from endophytes not only overcomes the problems of the slow growth cycle and low yield of medicinal plants, but also significantly enriches pesticide resources [17,18]. So far, only one article has reported the use endophytic bacteria of *W. indica* to synthesize silver nanoparticles against human pathogenic bacteria and fungi [37]. Therefore, developing and using endophytic bacteria to synthesize target compounds has important application value. We selected the most active endophyte, *P. aeruginosa* W-126, from 209 endophytes of *W. indica* through observation of the anti-bacterial wilt zone. Based on active compound tracking principles, the compound PCN was isolated from dozens of components of endophyte W-126.

Although several studies have reported that *P. aeruginosa* could be used as a biocontrol bacterium to prevent bacterial wilt [22,23,24] and root-knot nematode disease [12,25,26], only a few active compounds, such as acridine-4-carboxylic acid, have been isolated from the fermented broth of *P. aeruginosa* B27 and found to show certain nematicidal activity against *M. incognita* [38]. What is more, in addition to degrading enzymes [28,39], whether *P. aeruginosa* can produce other small active compounds to inhibit *R. solanacearum* is rarely reported. Phenazine and its derivatives, as antibacterial agents, could be produced by *P. aeruginosa* [29,30,40]. PCN could trigger the innate immune response in *C. elegans*. The nuclear hormone receptor NHR-86/HNF4 was validated as a PCN sensor in *C. elegans*, and PCN bound to the ligand-binding domain of NHR-86/HNF4 [41]. However, the activities of PCN and other phenazine compounds against bacterial wilt and *M. incognita* are rarely reported. Therefore, it is of great significance to study the structure–activity relationship of phenazine compounds against *R. solanacearum* and *M. incognita*.

When the concentration of PCN analogues was 125 μg/mL, the inhibitory rates of PCN and phenazine against *R. solanacearum* were over 60% (Table 2). The inhibitory rate of phenazine against *R. solanacearum* was 66.33%; although PCN was not the most active of these compounds, it showed the second highest inhibitory rate of all the PCN analogues, with 68.18% (Table 2). These results indicate that PCN and phenazine had excellent inhibitory activity against *R. solanacearum*. In addition, the inhibitory rate of phenazine-1-carboxylic acid (PCA), which is produced by *P. aeruginosa*, was only 21.02%. The different activity results indicate that the acylamide group increased the activity of phenazine against *R. solanacearum*, while the carboxylic acid group decreased its activity. However, no work was carried out on the mechanism of PCN and phenazine against *R. solanacearum*. The molecular docking analysis indicated that PCN bound to one of the candidate target proteins, pyruvate dehydrogenase [42]. PCN inhibited the activity of histone acetyltransferase Gcn5 [43]. Other studies reported that PCN bound to the histone acetyltransferase (HAT) domain of FgGcn5 at its cosubstrate acetyl-CoA binding site, thus competitively inhibiting the histone acetylation function of the enzyme [44]. Certain reports support the potential of halogenated phenazine compounds as antibiofilm agents and provide a foundation for rational drug design targeting the AgrA-DNA interaction [45]. Reports also support the bioactive naphthoquinone and phenazine analogs from endophytic *Streptomyces* sp. PH9030 as α-Glucosidase inhibitors [46]. In addition, the literature reports that phenazine could act as a dimer-disrupting inhibitor of TcTIM, which is a glycolytic enzyme essential for parasite survival [47]. However, more work needs to be carried out to clarify whether PCN acts on these targets and where PCN inhibits these sites in *R. solanacearum* and *M. incognita*.

In a pot experiment of PCN regarding the control of the complex diseases of *R. solanacearum* and *M. incognita*, PCN, streptomycin sulfate, W-126, and Zhuorun all showed good control effects against bacterial wilt disease. The control efficiencies of PCN and streptomycin sulfate were, respectively, 51.41 and 57.76%, which were significantly higher than the biocontrol agent of W-126 and Zhuorun, with 37.57 and 30.58%, respectively (Table 3, Appendix A). These results indicate that PCN shows similar control efficiency against bacterial wilt disease with streptomycin sulfate when the complex diseases of *R. solanacearum* and *M. incognita* occur. Moreover, the control efficiencies of these two compounds were higher than the biocontrol agent of Zhuorun^®^. Zhuorun^®^ is the commercial biocontrol agent of *B. amyloliquefaciens* QST713 in China, and is similar to the international commercial bactericide Serende^®^. The control efficiencies of Zhuorun^®^ against *R. solanacearum* and *M. incognita* were, respectively, 30.58 and 60.16%. However, the control efficiencies of PCN were, respectively, 51.41 and 39.80%. This is the first report to systematically separate and use PCN to control bacterial wilt and root-knot nematode diseases, providing a reference for how to use biocontrol agents to control root-knot nematode complex diseases.

## 4. Materials and Methods

### 4.1. Bacterial Cultures and Nematode Inoculum

#### Plant Materials and Test Strains

Biocontrol strain: In total, 209 endophytic bacteria were collected from *Waltheria indica* L. (Sterculiaceae) from 31 collection sites in Hainan province, China. All the bacteria were genetically stable single species, and all of them were grown on LB media at 28 °C and preserved in the School of Tropical Agriculture and Forestry, Hainan University, China.

Plant pathogenic strains: *Ralstonia solanacearum*, *Xanthomonas oryzae* pv. oryzae, *Xanthomonas manihotis*, *Pseudomonas syringae* DC3000, *Xanthomonas oryzae* pv. oryzicola, and *Xanthomonas campestris* pv. citri (Hasse) Dye were cultivated on LB media at 28 °C and preserved in the School of Tropical Agriculture and Forestry, Hainan University, China.

Cultivation of *M. incognita*: All the nematodes were obtained from pepper roots infected by *M. incognita*. The pepper roots were rinsed, the eggs were picked from the pepper roots, and then the eggs were incubated in a self-made incubator. The peppers infected by *M. incognita* were cultured in the Bioassay Center of the School of Tropical Agriculture and Forestry, Hainan University.

Chemicals: All organic solvents used in the study were of analytical grade and purchased from Xilong Chemical Co., Ltd. (Shanghai, China). All chemical reagents were purchased from Innochem (Beijing, China) or Bide Pharmatech Co., Ltd. (Shanghai, China).

### 4.2. Antibacterial Activity Determination of Endophytic Bacteria Against R. solanacearum

#### 4.2.1. Preparation of *R. solanacearum* Suspension

*R. solanacearum* was activated on LB plates, and single colonies were placed into 50 mL conical bottles containing LB liquid medium on a super-clean lab benchtop. Then, the conical bottle was shaken overnight at 28 °C and 125 r/min. The *R. solanacearum* suspension was diluted with an OD_600_ value of 1.0, and subsequently stored at 4 °C for future use.

#### 4.2.2. Antibacterial Activity Determined by the Filter Paper Diffusion Method

According to the method of He [48], the *R. solanacearum* suspension was added to the LB medium at 1%, and then the LB medium with *R. solanacearum* suspension was poured into a Petri dish for use. After that, two pieces of 6 mm filter paper dried by high-pressure steam sterilization were placed 2 cm away from the center point of the Petri dish. Then, 10 μL of biocontrol bacteria suspension was added to each filter paper. Filter paper with 10 μL LB solution was used as a blank control (CK), and filter paper with 10 μL streptomycin (20 μg/mL) was used as a positive control (CK+). Three petri dishes were repeated for each treatment. The Petri dishes were placed in an artificial climate chamber at (28 ± 1) °C for 72 h. When the *R. solanacearum* colony of CK filled the whole dishes, the size of the bacteriostatic zones was measured with a ruler.

### 4.3. Identification of Strain W-126

Colony morphology, physiological, and biochemical identification was undertaken according to the steps in the Manual of Common Bacterial System Identification. The strain W-126 was cultured on the surface of the LB solid medium with the standard dilution plate method [49], and the morphological characteristics of the colonies were observed after the single colony of W-126 grew out. The activated single colony was selected for Gram staining, and the microscopic morphological characteristics of the strain were observed under the optical microscope after 24−48 h of culture.

A bacterial DNA extraction kit was used to extract the genomic DNA of strain W-126, and PCR amplification was performed using 27F and 1492R as amplification primer templates [25]. The PCR amplification products were sequenced by Sangon Biotech (Shanghai) Co., Ltd. The sequence results were analyzed and compared with other sequences in the NCBI database. The phylogenetic tree of W-126 was constructed using MEGA11.0 software. The phylogenetic tree was based on 16S rDNA sequences, and sequence similarity analysis was employed to elucidate the evolutionary relationships among the organisms under study [48].

### 4.4. Isolation and Extraction of Bacteriostatic Substances from Strain W-126

#### 4.4.1. Crude Extraction and Its Bacteriostatic Activity Determination

The organic solvent extraction method was used to extract bacteriostatic substances. The biocontrol strain W-126 was cultured for 3 days in LB liquid medium at 28 °C, 125 r/min, and the supernatant was taken for extraction. W-126 supernatant was extracted with petroleum ether, ethyl acetate, or n-butanol, and the obtained organic solvent was then spin-steamed to obtain crude extract. The crude extracts of each organic phase (2 mg) were dissolved in 200 μL methanol solution and then diluted with sterile water into 1 mg/mL liquors. The antibacterial rate of the liquors was evaluated according to the filter paper diffusion method mentioned in Section 4.2.2. The size of the antibacterial zone was measured with a ruler.

#### 4.4.2. Isolation of Bacteriostatic Substances from Strain W-126 Crude Extraction

The ethyl acetate phase extraction of W-126 supernatant was separated and purified by silica gel column chromatography, and seven components (Fr.1, Fr.2, Fr.3, Fr.4, Fr.5, Fr.6, and Fr.7) were separated by petroleum ether/ethyl acetate gradient elution (1:0, 4:1, 3:2, 1:1, 2:3, 1:4, and 0:1). Using Sephadex LH-20 column chromatography and methanol for elution, the bacteriostatic component Fr.3 was further purified, and 11 components were obtained (Fr.3.1, Fr.3.2, Fr.3.3, Fr.3.4, Fr3.5, Fr.3.6, Fr.3.7, Fr.3.8, Fr.3.9, Fr.3.10, and Fr.3.11). A semi-preparative liquid chromatograph (Cosmosil, 5C18-MS-II, 20ID × 250 mm, flow rate 5 mL/min; the absorption wavelength of the ultraviolet detector was 252 nm; mobile phase: acetonitrile/water, 60:40, *v*/*v*) was used to further purify the bacteriostatic component Fr.3.5, and six components (Fr.3.5.1, Fr.3.5.2, Fr.3.5.3, Fr.3.5.4, Fr.3.5.5, and fr.3.5.6) were obtained. Finally, the antibacterial component of Fr.3.5.2 was determined as the antibacterial compound using the filter paper diffusion method.

### 4.5. Identification of Bacteriostatic Compound from Strain W-126

The chemical structure of the active compound was determined via nuclear magnetic resonance (NMR) and high-resolution mass spectrometry (HRMS). NMR spectra were obtained for Chloroform-*d* on an Avance Neo 400 spectrometer (Bruker BioSpin Corp., Karlsruhe, Germany). HRMS data were obtained on an Agilent 6210 ESI/TOF MS (Agilent Technologies, Inc., Santa Clara, CA, USA). The mobile phase comprised acetonitrile/water at a ratio of 60:40. The mass spectrometry test method was as follows: ESI ion mode was used for mass spectrometry detection, and the scanning range as 100~500 *m*/*z*.

### 4.6. Antibacterial Determination of PCN Against 6 Phytopathogenic Bacteria and M. incognita

The EC_50_ of PCN against 6 phytopathogenic bacteria were determined by 96-well microculture plates based on the reported methods of Fan [50] and Mao [51]. The toxicity of the compound to *M. incognita* was determined by the immersion method [52]. PCN was dissolved in a small amount of DMF and methanol (ratio 1:100) to prepare a mother liquor with a concentration of 1000 μg/mL. The phytopathogenic bacteria, cultured overnight, were adjusted to a solution with a concentration of 1 × 10^6^ cfu/mL for use. The 100 μL bacterial solution and 100 μL diluted PCN solution were added to a 96-well culture plate, and the PCN concentrations were dissolved to 250, 125, 62.5, 31.25, 15.63, 7.81, 3.91, and 1.95 μg/mL. Streptomycin with a concentration of 125 μg/mL was used as the positive control (CK+) and the DMF and methanol solution was used as the blank control (CK). The total amount of each well was 200 μL, and each treatment was repeated 4 times. The 96-well plates were cultured in a biochemical incubator at 28 °C for 24 h, and the growth of *R. solanacearum* and other pathogenic bacteria were detected by ELIASA at 600 nm. The inhibition rates of PCN against 6 phytopathogenic bacteria were calculated based on the measured OD values. The inhibition rate was calculated using the equation in [53], where the turbidity value of the blank control group is expressed as X and the turbidity value of the treatment group is expressed as Y. Inhibition rate (%) = (X − Y) × 100/X. The mortality and correct mortality rates of *M. incognita* were calculated according to the following formulas [52]:Mortality rate (%) = (Number of dead nematodes)/(Total nematodes) × 100Corrected mortality rate (%) = (Treatment group mortality − Control group mortality)/(1 − Control group mortality) × 100

### 4.7. Activity Determination of PCN Analogues Against R. solanacearum and M. incognita

The activity of PCN analogues on *R. solanacearum* and *M. incognita* was measured via the method in Section 4.6. The nematode suspension for testing was prepared to a concentration of 100 ± 10 nematodes per milliliter using sterile water. The PCN analogues were respectively weighed to 10 mg, dissolved by adding 10 μL of DMF, and then diluted to 5 mL with 0.5% Tween-80 aqueous solution to obtain the mother compound suspension with a concentration of 2000 μg/mL. The compound suspensions were mixed with nematodes or bacteria suspension, and all the compound concentrations were set to 125 μg/mL. The concentration with 10 μg/mL of abamectin was used as a positive control against *M. incognita*, the concentration with 125 μg/mL of streptomycin was used as a positive control against *R. solanacearum*, and the 0.5% Tween-80 aqueous solution was used as a blank control (CK). Each treatment was repeated 3 times. After 24 h of treatment, the mortality and correct mortality rates of nematodes were calculated as per Section 4.6. The growth of *R. solanacearum* was detected by ELIASA at 600 nm.

### 4.8. Pot Experiment on PCN for Controlling Complex Diseases of R. solanacearum and M. incognita

The experiment was divided into seven treatment groups: pepper with neither *R. solanacearum* nor *M. incognita* as the negative control (CK-), pepper with both *R. solanacearum* and *M. incognita* as the blank control (CK), pepper with both *R. solanacearum* and *M. incognita* under PCN control (PCN), pepper with both *R. solanacearum* and *M. incognita* under the treatment of W-126 fermentation broth (W-126), pepper with both *R. solanacearum* and *M. incognita* under Zhuorun treatment (Zhuorun) as a positive control against the complex diseases of *R. solanacearum* and *M. incognita*, pepper with both *R. solanacearum* and *M. incognita* under abamectin treatment (Abamectin) as a positive control against *M. incognita,* pepper with both *R. solanacearum* and *M. incognita* under streptomycin sulfate treatment (streptomycin sulfate) as a positive control against *R. solanacearum*. Each treatment had 7 pepper plants. Pepper seedlings were transplanted in a seedling bowl (10 cm in diameter and 15 cm in height) for 1 month. Then, pepper seedlings were pulled with 5 mL PCN or streptomycin sulfate suspension with a concentration of 100 μg/mL. Abamectin suspension was pulled with 50 mL at a concentration of 10 μg/mL. Biocontrol bacterium W-126 was pulled with 50 mL at a concentration of 10^7^ CFU/mL (or 50 mL solution of *Bacillus amyloliquefaciens* QST713 with a concentration of about 10^7^ CFU/mL, purchased from Bayer (named Zhuorun^®^); https://www.bayer.com.cn/zh-hans/zhuorun-0, accessed on 24 May 2023). *B. amyloliquefaciens* QST713 was propagated and reported with ability against *R. solanacearum* and *M. incognita* [54,55]. On the second day, pepper seedlings were inoculated with 600 ± 10 nematodes. On the third day, each plant was inoculated with a 10 mL strain suspension of *R. solanacearum* with a concentration of 1 × 10^7^ CFU/mL. The number of root knots and the disease index of bacterial wilt were investigated 35 days after nematode inoculation. The disease index of bacterial wilt and the control effect of bacterial wilt and nematodes were calculated as per previous studies [52,56].

### 4.9. Statistical Analysis

The data were analyzed using SPSS version 27.0 software via a one-way analysis of variance (ANOVA) followed by a Tukey test. All the data are presented as mean ± SE. *p* < 0.05 was regarded as statistically significant. The figures were made with GraphPad Prism software 9.0 or Photoshop software 2023.

## 5. Conclusions

*Pseudomonas aeruginosa* W-126 was isolated from 209 endophytic bacteria of *W. indica* with an efficient antagonistic effect against *R. solanacearum*. Based on active compound tracking principles, phenazine-1-carboxamide (PCN) was isolated and identified from *P. aeruginosa* W-126. PCN showed certain inhibitory activity against six phytopathogenic bacteria. It displayed excellent inhibitory activity against *R. solanacearum*, with an EC_50_ of 64.16 μg/mL. In addition, PCN showed certain nematocide activity, with an LC_50_ value of 118.63 μg/mL at 72 h. The structure−activity relationship indicated that the phenazine skeleton and acylamide groups were the key pharmacophores for the activity of phenazine-related compounds against *R. solanacearum*. PCN inhibited the complex diseases of *R. solanacearum* and *M. incognita* in a pot experiment. The inhibitory rates of *R. solanacearum* and *M. incognita* were 51.41 and 39.80%, respectively, when the concentration of PCN was 125 μg/mL.

## Figures and Tables

**Figure 1 ijms-26-03335-f001:**
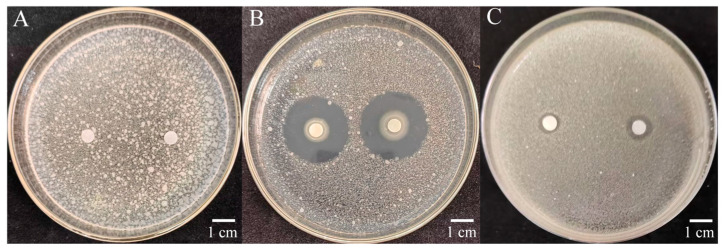
Inhibition effect of W-126 against *Ralstonia solanacearum.* (**A**): inhibition effect of LB solution against *R. solanacearum* (CK); (**B**): inhibition effect of strain W-126 against *R. solanacearum*; (**C**): inhibition effect of streptomycin against *R. solanacearum* (CK+).

**Figure 2 ijms-26-03335-f002:**
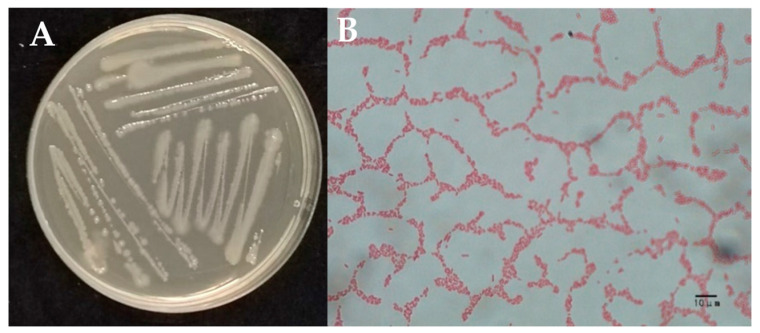
Morphological observation of strain W-126. (**A**): colony morphology of W-126 on LB plate; (**B**): bacterial Gram staining results.

**Figure 3 ijms-26-03335-f003:**
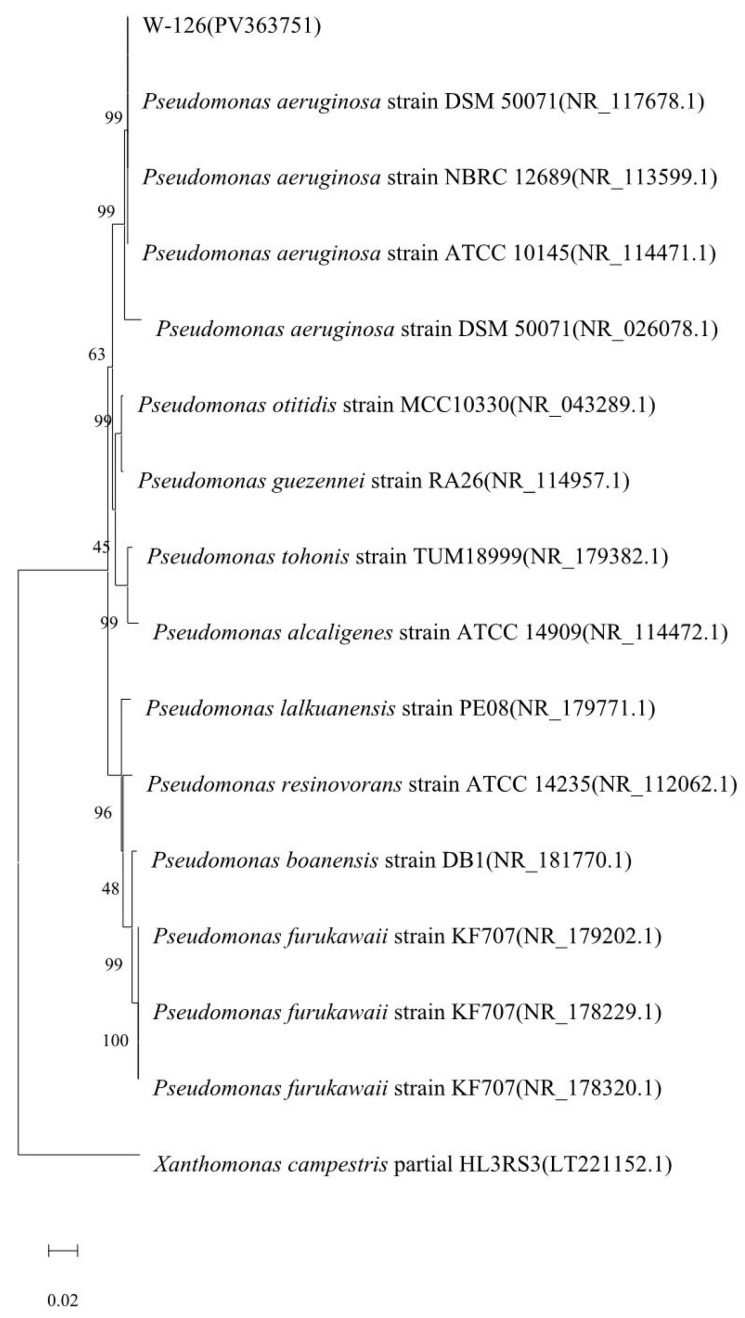
Phylogenetic evolution tree of strain W-126 based on 16S rDNA gene sequence.

**Figure 4 ijms-26-03335-f004:**
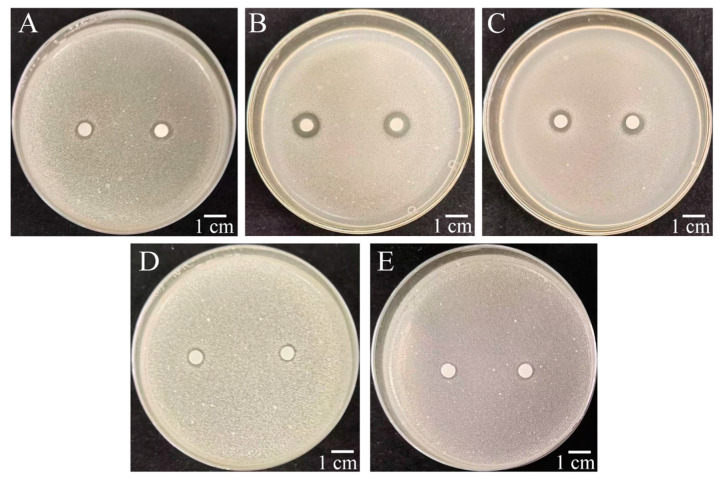
Antibacterial effect of organic solvent extracts of W-126 secondary metabolite. (**A**): petroleum ether phase; (**B**): ethyl acetate phase; (**C**): n-butanol phase; (**D**): methanol control; (**E**): control (blank control).

**Figure 5 ijms-26-03335-f005:**
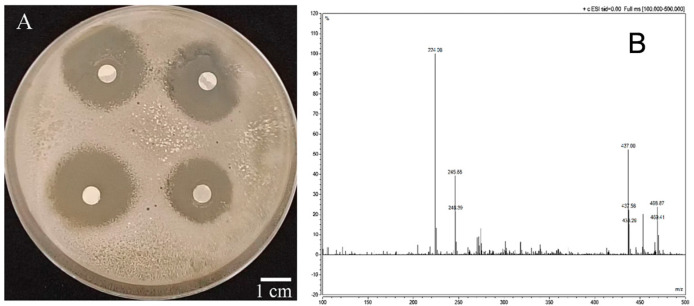
The inhibitory effect on *R. solanacearum* (**A**), mass spectrum (**B**), carbon spectrum (**C**), and hydrogen spectrum (**D**) of compound W-s1.

**Figure 6 ijms-26-03335-f006:**
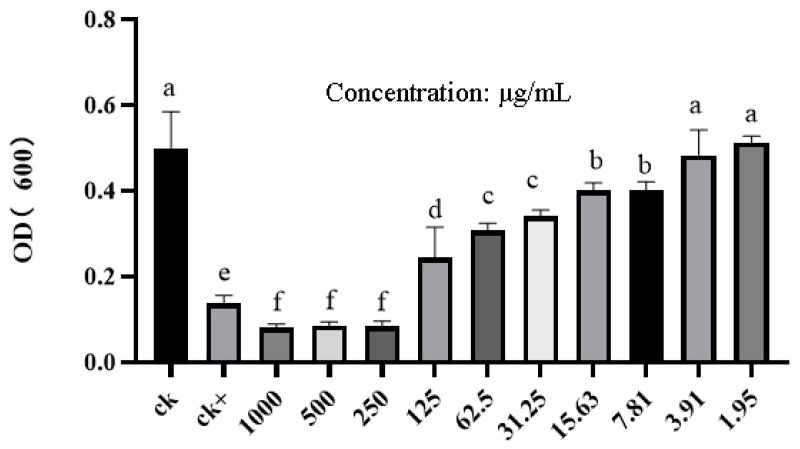
Inhibition activity of PCN at different concentrations against *R. solanacearum*. ck: *R. solanacearum* suspension with 0.5% Tween-80 aqueous solution; ck+: *R. solanacearum* suspension in streptomycin at a concentration of 125 μg/mL. The same lower-case letter on the column means no significant difference in a Tukey test (*p* < 0.05).

**Table 1 ijms-26-03335-t001:** EC_50_ of PCN against 6 phytopathogenic bacteria and LC_50_ against *M. incognita*.

Pathogenic Bacteria	Regression Equation	R^2^	EC_50_ (µg/mL)	Inhibition Rate (%) at 250 µg/mL
*X. oryzae* pv. oryzae	Y = −4.17 + 1.86x	0.915	184.10	54.25 ± 0.39
*X. manihotis*	Y = −12.59 + 4.95x	0.997	361.71	31.53 ± 0.75
*P. syringae*	Y = −8.35 + 3.39x	0.999	286.56	45.23 ± 0.53
*X. oryzae* pv. oryzicola	Y = −5.35 + 2.06x	0.989	387.29	40.86 ± 1.03
*X. campestris*	Y = −8.4 + 3.93x	0.991	138.70	71.46 ± 0.12
*R. solanacearum*	Y = −4.76 + 2.66x	0.950	64.16	87.88 ± 0.99
*M. incognita*	Y = −9.57 + 4.63x	0.975	118.63	83.68 ± 2.17

The statistical analysis was conducted by the ANOVA method under the conditions of equal variances assumed (*p* > 0.05) and equal variances not assumed (*p* < 0.05).

**Table 2 ijms-26-03335-t002:** Inhibitory effects of PCN analogues and positive control on *R. solanacearum* and *M. incognita*.

Number	Compounds	CAS	Inhibitory Rate (%)	
			*R. solanacearum*	*M. incognita*
1	Phenazin-1-ylamine	2876-22-4	−7.63 ± 1.70	66.86 ± 1.50
2	3-Amino-phenazin-2-ol	4569-77-1	63.41 ± 0.75	25.58 ± 0.89
3	1-Methoxyphenazine	2876-17-7	−19.85 ± 4.42	34.29 ± 1.58
4	PCN	550-89-0	68.18 ± 2.15	54.62 ± 2.29
5	Phenazine	92-82-0	66.33 ± 5.81	52.52 ± 1.92
6	1-Hydroxyphenazine	528-71-2	33.34 ± 1.37	62.67 ± 1.97
7	Dipyrido [3,2-a:2′,3′-c]phenazine	19535-47-8	9.12 ± 1.92	23.65 ± 2.12
8	Phenazine-1-carboxylic acid	2538-68-3	21.02 ± 1.61	12.41 ± 1.34
9	Safranine T	477-73-6	−34.17 ± 7.16	28.57 ± 0.74
10	Phenazine methosulfate	299-11-6	82.77 ± 4.73	16.67 ± 0.58
11	Neutral Red	553-24-2	67.00 ± 1.07	9.67 ± 0.58
12	2,3-Diaminophenazine	655-86-7	−24.48 ± 6.9	19.66 ± 2.05
13	Azocarmine G	25641-18-3	−25.89 ± 2.67	6.97 ± 0.64
14	Methylene Violet 3RAX	4569-86-2	30.02 ± 7.77	27.40 ± 0.81
15	Phenazine ethosulfate	10510-77-7	37.06 ± 2.10	8.56 ± 0.63
16	1,6-Phenazinediol	69-48-7	44.04 ± 2.07	15.25 ± 0.79
17	Phenazine 5,10-dioxide	303-83-3	14.55 ± 0.80	22.00 ± 0.91
18	Abamectin	71751-41-2	-	100
19	Streptomycin sulfate	3810-74-0	98.91 ± 1.48	-

Values are expressed as means ± SE.

**Table 3 ijms-26-03335-t003:** Pot experiment results for PCN and control treatments against complex diseases of *R. solanacearum* and *M. incognita*.

Treatment	Plant Height (cm)	Plant Weight (g)	*M. incognita*		*R. solanacearum*	
			Root-Knot Number	Control Efficiency (%)	Disease Index	Control Efficiency (%)
CK-	35.09 ± 2.16a	2.80 ± 0.12a	-	-	-	-
CK	34.26 ± 1.84a	2.22 ± 0.15b	68.67 ± 1.53a	-	73.52 ± 3.83a	-
PCN	34.00 ± 2.76a	2.38 ± 0.19b	41.33 ± 0.58b	39.80 ± 0.90c	35.80 ± 4.28d	51.41 ± 3.35a
Zhuorun	34.03 ± 2.83a	2.38 ± 0.10b	27.33 ± 1.15d	60.16 ± 2.41a	50.62 ± 4.28c	30.58 ± 1.74b
W-126	33.75 ± 1.83a	2.36 ± 0.09b	31.33 ± 2.08c	54.31 ± 3.96b	45.68 ± 4.28c	37.57 ± 8.76b
Streptomycin sulfate	34.50 ± 2.93a	2.35 ± 0.06bc	66.67 ± 1.53a	2.92 ± 0.07d	30.86 ± 4.27d	57.76 ± 7.74a
Abamectin	32.38 ± 3.58a	2.34 ± 0.07bc	23.33 ± 2.08e	65.99 ± 3.45a	65.43 ± 4.28b	11.04 ± 1.82c

Values are expressed as means ± SE. The means in the same column followed by the same lower-case letter were not significantly different in a Tukey test (*p* < 0.05).

## Data Availability

Data are contained within the article and Appendix A.

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
