# Peer review of "Integrated Management of Bacterial Wilt and Root-Knot Nematode Diseases in Pepper: Discovery of Phenazine-1-Carboxamide from *Pseudomonas aeruginosa* W-126"

_ijms, 2025, doi:10.3390/ijms26073335_

Round 1
Reviewer 1 Report (New Reviewer)
Comments and Suggestions for Authors
The authors screened a biocontrol bacterium W-126 which was highly effective in inhibiting R. solanacearum from 209 endophytic bacteria of Waltheria indica L. Based on active compound tracking principles, active compound PCN was isolated and Identified with TLC analysis. The PCN was did antibacterial determination against 6 phytopathogenic bacteria and activty against M. incognita. Activities of 17 PCN analogues against R. solanacearum and M. incognita were also determinated. Finally evaluated the PCN on controlling complex disease of R. solanacearum and M. incognita.
The introduction is well written, with appropriate use of bibliographical sources. The M&Ms are quite rigorous, conducted with the right equipment and the data analysis is sufficient. I have following suggestions:
1.LINE 111-114 provide the accession no of 16S of W-126 .
- Line 134-I think this place should be misdescribed. It should be Figure 5A.
- Line 169- “the inhibitory rate”should be“the inhibitory rates”.
- Line 193- “The control efficiency of”should be “The control efficiencies of”.
- Line 193- I think this place should bemisspelled the word of “treaments”.
- Combine fig.5 and 6.
- Delete Fig. 7
Author Response
comments1: LINE 111-114 provide the accession no of 16S of W-126 .
response1: Thank you for pointing this out, we have applied in NCBI for the accession no, and we have added the accession no in Figure 3.
comments2: Line 134-I think this place should be misdescribed. It should be Figure 5A.
response2: Thank you for pointing this out, this sentence does not accurately describe the result of the picture, we have revised the Figure 5 and the description of it.
comments3: Line 169- “the inhibitory rate”should be“the inhibitory rates.
response3: Thank you for pointing this out, we have revised the description in section 2.6.
comments4: Line 193- “The control efficiency of”should be “The control efficiencies of”.
response4: Thank you for pointing this out, we have revised the description.
comments5: Line 193- I think this place should be misspelled the word of “treaments”.
response5: Thank you for pointing this out, we have revised the word in the title of Table 3.
comments6: Combine fig.5 and 6.
response6: Thank you for pointing this out, we have combined fig.5 and 6 together.
comments7: Delete Fig. 7
response7: Thank you for pointing this out, other reviewers asked us to modify Figure 7, so we kept Figure 7 and modified it.
Reviewer 2 Report (New Reviewer)
Comments and Suggestions for Authors
For several of the figures, please explain the reason for adding a red line that has no text attached to it?
In the figure and in some of the following figures, a space must be given to the text that follows.
For figure 3, the EBI accession numbers begin with two letters followed by an underscore and then the 6 numbers follow.
Figure 7, remove the space between the parentheses and the attached text ( Concentration:
Line 222 remove the following repeated words "is a"
Author Response
Comments 1: For several of the figures, please explain the reason for adding a red line that has no text attached to it?
Response 1: Thank you for pointing this out, Our article has been reviewed once, the current version is re-submission. The red line on the picture meaning of the deleted old pictures, we have updated new pictures in it.
Comments 2: In the figure and in some of the following figures, a space must be given to the text that follows.
Response 2: Thank you for pointing this out, spaces have been given to the text that following figures.
Comments 3: For figure 3, the EBI accession numbers begin with two letters followed by an underscore and then the 6 numbers follow.
Response 3: Thank you for pointing this out, we have added an underscore between the EBI accession numbers and the two letters.
Comments 4: Figure 7, remove the space between the parentheses and the attached text ( Concentration:
Response 4: Thank you for pointing this out, we have changed the Figure 7.
Comments 5: Line 222 remove the following repeated words "is a"
Response 5: Thank you for pointing this out, we have deleted one of "is a".
Reviewer 3 Report (New Reviewer)
Comments and Suggestions for Authors
line 93 - "antibacterial zone" should be changed to "inhibition zone"
Fig. 1 - what means CK, CK+ in the fig. description? and what is this the red line on the picture?
line 122 - "antibacterial circles appeared" should be "inhibition zones appeared"
line 125 - "inhibitory circle" should be "inhibitory zone"
line 135 - "the antibacterial diameter reached 25 mm" should be "the inhibition zone diameter reached 25 mm"
line 136 - "antibacterial zone" should be "inhibition zone"
line 160-161 - Fig. 7 description contains ck+. What it means? Is it the same as CK+ mentioned above?
Symbols CK/CK+ have to be explained when were used for the first time in Results.
line 185, 418, and others - (Zhuorun) ... explain in line 188 what it means?, what is this? Reading results I don't know what is this Zhuorun.
line 222 - twice " is a", remove one
line 240 - ".... ]. Only a few active compounds ..." seems to be a continuation of the previous sentence; thus there should be "....], only a few active compounds.... "
line 275 - "....incognita. Both the PCN, streptomycin sulfate ..." should be ".... incognita, both the PCN, streptomycin sulfate..."
line 277 - "were respectively with 51.41, and 57.76%" - delete "with"
line 294 - "All the bacteria were genetic stability single species, and all of them ...." should be "All the bacteria were genetically stable single species, and all of them..."
Author Response
Comments 1: line 93 - "antibacterial zone" should be changed to "inhibition zone"
Response 1: Thank you for pointing this out, we have changed as "inhibition zone".
Comments 2: Fig. 1 - what means CK, CK+ in the fig. description? and what is this the red line on the picture?
Response 2: Thank you for pointing this out, we have added the description of CK, CK+ in the Fig. 1. Our article has been reviewed once, the current version is re-submission. The red line on the picture meaning of the deleted old pictures, we have updated new pictures in it.
Comments 3: line 122 - "antibacterial circles appeared" should be "inhibition zones appeared".
Response 3: Thank you for pointing this out, we have changed as "inhibition zones appeared".
Comments 4: line 125 - "inhibitory circle" should be "inhibitory zone".
Response 4: Thank you for pointing this out, we have changed as "inhibitory zone".
Comments 5: line 135 - "the antibacterial diameter reached 25 mm" should be "the inhibition zone diameter reached 25 mm".
Response 5: Thank you for pointing this out, we have changed as "the inhibition zone diameter reached 25 mm".
Comments 6: line 136 - "antibacterial zone" should be "inhibition zone".
Response 6: Thank you for pointing this out, we have changed as "inhibition zone".
Comments 7: line 160-161 - Fig. 7 description contains ck+. What it means? Is it the same as CK+ mentioned above?
Response 7: Thank you for pointing this out, ck+ means R. solanacearum suspension in streptomycin with concentration of 125 μg/mL. We have added the description of ck and ck+.
Comments 8: Symbols CK/CK+ have to be explained when were used for the first time in Results.
Response 8: Thank you for pointing this out, we have added the description of ck and ck+ in the Figure 7, we have described them in the section of 4.6.
Comments 9: line 185, 418, and others - (Zhuorun) ... explain in line 188 what it means?, what is this? Reading results I don't know what is this Zhuorun.
Response 9: Thank you for pointing this out, we have described them in the line 182-183, line 278-279 and line424-426.
Comments 10: line 222 - twice " is a", remove one
Response 10: Thank you for pointing this out, we have deleted one of " is a".
Comments 11: line 240 - ".... ]. Only a few active compounds ..." seems to be a continuation of the previous sentence; thus there should be "....], only a few active compounds.... "
Response 11: Thank you for pointing this out, we have we have changed the description.
Comments 12: line 275 - "....incognita. Both the PCN, streptomycin sulfate ..." should be ".... incognita, both the PCN, streptomycin sulfate..."
Response 12: Thank you for pointing this out, we have we have changed the description.
Comments 13: line 277 - "were respectively with 51.41, and 57.76%" - delete "with"
Response 13: Thank you for pointing this out, we have we have deleted "with".
Comments 14: line 294 - "All the bacteria were genetic stability single species, and all of them ...." should be "All the bacteria were genetically stable single species, and all of them..."
Response 14: Thank you for pointing this out, we have we have changed the description.
This manuscript is a resubmission of an earlier submission. The following is a list of the peer review reports and author responses from that submission.
Round 1
Reviewer 1 Report
Comments and Suggestions for Authors
There is continued interest in your manuscript titled “Integrated Management of Bacterial Wilt and Root-Knot Nematode Diseases in Pepper: Discovery of Phenazine-1-carbox-amide from Pseudomonas aeruginosa W-126” which you submitted to the IJMS. Bacterial wilt, caused by Ralstonia solanacearum, is one of the most serious diseases affecting pepper (Capsicum annuum L.). The concurrent infection of R. solanacearum and root-knot nematodes (Meloidogyne spp.) exacerbates the severity of bacterial wilt in pepper. Utilizing plant endophytic bacteria to control these mixed diseases is a viable strategy. Your MS has been conditionally accepted. Please carefully revise the following content according to review comments.
1. Please add a scale bar in Figure 1, Figure 4, and Figure 5.
2. Please italicize the 'p' in the notes of Table 1, Table 2, and Table 3.
3. Please provide physiological and molecular biological evidence for the prevention and treatment of pepper wilt and root knot nematode disease using W-126.
Comments on the Quality of English LanguageThe English of your manuscript must be improved before resubmission. We strongly suggest that you obtain assistance from a colleague who is well-versed in English or whose native language is English.
Author Response
Comments1: Please add a scale bar in Figure 1, Figure 4, and Figure 5.
Thank you for pointing this out, we have added a scale bar in Figure 1, Figure 4, and Figure 5.
Comments2: Please italicize the 'p' in the notes of Table 1, Table 2, and Table 3.
Thank you for pointing this out, I have italicized the 'p' in the notes of Table 1, Table 2, and Table 3.
Comments3: Please provide physiological and molecular biological evidence for the prevention and treatment of pepper wilt and root knot nematode disease using W-126.
Thank you for pointing this out, I have provided physiological and molecular biological evidence for the prevention and treatment of pepper wilt and root knot nematode disease using phenazine-1-carboxamide (PCN) produced by W-126 in the Supplementary Figure S1. We did not want to use strain W-126 to prevent pepper wilt and root knot nematode disease. Because strain W-126 was Pseudomonas aeruginosa, and P. aeruginosa is one nosocomial opportunistic pathogen.
4. Comments on the Quality of English Language
The English of your manuscript must be improved before resubmission. We strongly suggest that you obtain assistance from a colleague who is well-versed in English or whose native language is English.
Thank you for pointing this out, I have asked a foreign teacher in our college, Farooq Saqib check and amend our manuscript. And I added his name in the Acknowledgments.
Reviewer 2 Report
Comments and Suggestions for Authors
The paper entitled “Integrated Management of Bacterial Wilt and Root-Knot Nematode Diseases in Pepper: Discovery of Phenazine-1- carbox-amide from Pseudomonas aeruginosa W-126” it is a well written work. The introduction is well written, with appropriate use of bibliographical sources. The M&Ms are quite rigorous (but with important omission), conducted with the right equipment and the data analysis is sufficient. The results are consistent with M&Ms. The discussion is not speculative.
Nonetheless, some fundamental points remain unclear. The authors themselves declare that the antibacterial capabilities of PCN are known (L69), but there is little data for bacterial wilt. At this point, we would expect an in vitro and in vivo evaluation of PCN (perhaps compared with other well-known molecules effective against bacterial wilt, but just one comparison treatment was carried out with a so called “Zhuorun”, a compound unknown to me of which there is no trace even on the internet, not even on the Bayer website). Instead, the authors carry out research that involves the identification of PCN from an endophyte taken from a medicinal plant. What is the usefulness of this investigation? We already know that P. aeruginosa produces PCN, why understand if the W-126 strain does? It is quite obvious that it produces it, in case it would be interesting to know if it produces PCN in greater quantities than other strains. Otherwise, it would have made much more sense to identify, in a new strain, an alternative molecule to PCN.
L9: bacteria wilt is not always caused by Ralstonia solanacearum, so rephrase the sentence (eg. Bacterial wilt of pepper (C. annuum) caused by…)
L18: please avoid the use of “modern”
L19-20: I think that an “in vitro” detail is needed.
L24: some numbers will help the reader to understand the impact of inhibition. Furthermore, is better to write about “control” instead of “inhibition” for field trials.
L31: as above, please be more specific about bacterial wilt.
L35: at worldwide level?
L38: please delete “vegetable”.
L167-L341: “Zhuorun” is a compound unknown to me of which there is no trace even on the internet, not even on the Bayer website. I therefore have very serious doubts about the validity of this experiment. Furthermore, in the only text I found about it (https://www.mdpi.com/2073-4395/12/9/2177), it seems to be a product based on Bacillus amyloliquefaciens. So, even if "Zhuorun" existed (and I have many doubts), it would not make sense to compare a chemical molecule (PCN) treatment with a biological agent. If anything, it made more sense to compare a treatment with W-126 and "Zhuorun".
L222-225. It is a very generic statement, it is not reported the strain or how they are assessed for genetic correspondence.
L344: which bacillus?
L346: did you really start counting the nematodes for each inoculation?
L348-349: This cannot be reported without giving details of how it was done.
Author Response
Comments 1: Nonetheless, some fundamental points remain unclear. The authors themselves declare that the antibacterial capabilities of PCN are known (L69), but there is little data for bacterial wilt. At this point, we would expect an in vitro and in vivo evaluation of PCN (perhaps compared with other well-known molecules effective against bacterial wilt, but just one comparison treatment was carried out with a so called “Zhuorun”, a compound unknown to me of which there is no trace even on the internet, not even on the Bayer website). Instead, the authors carry out research that involves the identification of PCN from an endophyte taken from a medicinal plant. What is the usefulness of this investigation? We already know that P. aeruginosa produces PCN, why understand if the W-126 strain does? It is quite obvious that it produces it, in case it would be interesting to know if it produces PCN in greater quantities than other strains. Otherwise, it would have made much more sense to identify, in a new strain, an alternative molecule to PCN.
Thank you for pointing this out. To the first question, we described that the anti-bacterial wilt capability of PCN and other phenazine compounds, has been infrequently reported. So, we determined the activity of PCN analogues against R. solanacearum in vitro in the section of 4.7. Total 17 PCN analogues were tested, and the concentration of the PCN analogues was 125 μg/mL. Streptomycin with concentration of 125 μg/mL as positive control. The structure informations of the 17 PCN analogues were in the Supplementary Table S3. “Zhuorun” is a commercial biocontrol agent of Bacillus amyloliquefaciens QST713, which we used as a positive control in pot experiment to compare the control effect of PCN against R. solanacearum. For the second question, we did not simply identify PCN from an endophyte taken from a medicinal plant. We selected one endophyte W-126 from 209 biocontrol bacteria through observation the antibacterial zone of all the 209 bacteria. Based on active compound tracking principles, the compound PCN was isolated from endophyte W-126. For the third question, it would be more interesting to know if W-126 produces PCN in greater quantities than other strains.But the ability of W-126 produced PCN was normal. It would have made much more sense to identify, in a new strain, an alternative molecule to PCN. However, no the paper systematicly explored the anti-bacterial wilt capability of PCN and phenazine compounds. We did some work to explored the anti-bacterial wilt capability of PCN and phenazine compounds. We will also synthesize or purchase more PCN compounds to test their ability against bacterial wilt in the future.
Comments 2: L9: bacteria wilt is not always caused by Ralstonia solanacearum, so rephrase the sentence (eg. Bacterial wilt of pepper (C. annuum) caused by…)
Thank you for pointing this out. I agree with comment. Therefore, I have changed the description about it.
Comments 3: L18: please avoid the use of “modern”
Thank you for pointing this out. I have deleted the “modern” in this sentence.
Comments 4: L19-20: I think that an “in vitro” detail is needed.
Thank you for pointing this out. I have added the“in vitro”in this sentence.
Comments 5: L24: some numbers will help the reader to understand the impact of inhibition. Furthermore, is better to write about “control” instead of “inhibition” for field trials.
Thank you for pointing this out. I agree with comment. Therefore, I have added the control dates in the pot experiment. In addition, I have changed the “control” instead of “inhibition”in this sentence.
Comments 6: L31: as above, please be more specific about bacterial wilt.
Thank you for pointing this out. I agree with comment. Therefore, I have added specific description about bacterial wilt and two references.
Comments 7: L35: at worldwide level?
Thank you for pointing this out. Yes, I have added “ worldwide” in this sentence.
Comments 8: L38: please delete “vegetable”.
Thank you for pointing this out. I have deleted the “vegetable” in this sentence.
Comments 9: L167-L341: “Zhuorun” is a compound unknown to me of which there is no trace even on the internet, not even on the Bayer website. I therefore have very serious doubts about the validity of this experiment. Furthermore, in the only text I found about it (https://www.mdpi.com/2073-4395/12/9/2177), it seems to be a product based on Bacillus amyloliquefaciens. So, even if "Zhuorun" existed (and I have many doubts), it would not make sense to compare a chemical molecule (PCN) treatment with a biological agent. If anything, it made more sense to compare a treatment with W-126 and "Zhuorun".
Thank you for pointing this out. Yes, "Zhuorun" is the commercial biocontrol agent of Bacillus amyloliquefaciens QST713. I want to use chemical medicament as a positive control at the beginning, but I did not get the suitable medicine both against Ralstonia solanacearum and Meloidogyne incognita. B. amyloliquefaciens was reported with the ability aginst R. solanacearum and M. incognita. We just want to compare our compound with a commercial biocontrol agent. So, we finally chose "Zhuorun". It made more sense to compare a treatment with W-126 and "Zhuorun". But we did not want to use strain W-126 to prevent pepper wilt and root knot nematode disease. Because strain W-126 was Pseudomonas aeruginosa, and P. aeruginosa is one nosocomial opportunistic pathogen.
Comments 10: L222-225. It is a very generic statement, it is not reported the strain or how they are assessed for genetic correspondence.
Thank you for pointing this out. I agree with comment. Therefore, I have added specific description about the strain in this sentence paragraph.
Comments 11: L344: which bacillus?
Thank you for pointing this out. It was Bacillus amyloliquefaciens QST713, I have added it in this sentence.
Comments 12: L346: did you really start counting the nematodes for each inoculation?
Thank you for pointing this out. There will be some error, I have revised the description of this sentence.
Comments 13: L348-349: This cannot be reported without giving details of how it was done.
Thank you for pointing this out. I agree with comment. Therefore, I have added specific description about the details of control effect and two references in this part.
Reviewer 3 Report
Comments and Suggestions for Authors
I think the article is innovative and deserves publication. Nevetheless, some adjustments should be made, as follow:
Lines 31-33 - please give more data about the economic losses caused by bacterial wilt and the symptoms of the disease.
The Introduction does not describe Waltheria indica and the readers cannot understand why this plant was chosen for this experiment. Short description of Pseudomonas aeruginosa should be given as well.
Line 180 - the Discussion is not sufficient. There are a lot of experimental data but the discussion is lacking as well as the comparisons to similar works. Less than one page of DIscussion is too short for such paper.
Line 349 - the description of the software used in the study is missing.
Line 372 should be removed.
Line 385 - the References section is not made according to IJMS guidelines and should be corrected by the authors.
Author Response
Comments 1: Lines 31-33 - please give more data about the economic losses caused by bacterial wilt and the symptoms of the disease.
Response 1: Thank you for pointing this out. I agree with comment. Therefore, I have added specific description about bacterial wilt and two references.
Comments 2: The Introduction does not describe Waltheria indica and the readers cannot understand why this plant was chosen for this experiment. Short description of Pseudomonas aeruginosa should be given as well.
Response 2: Thank you for pointing this out. I agree with comment. Therefore, I have added specific description about Waltheria indica and three references.
Comments 3: Line 180 - the Discussion is not sufficient. There are a lot of experimental data but the discussion is lacking as well as the comparisons to similar works. Less than one page of DIscussion is too short for such paper.
Response 3: Thank you for pointing this out. I agree with comment. Therefore, I have added the discussion about comparison to similar works and specific description about structure information of PCN analogues with ablity against bacterial wilt and M. incognita.
Comments 4: Line 349 - the description of the software used in the study is missing.
Response 4: Thank you for pointing this out. I agree with comment. Therefore, I have added description of the software in section 4.9.
Comments 5: Line 372 should be removed.
Response 5: Thank you for pointing this out. I made a small mistake . I want to write “Data Availability Statement”, I have corrected the description here.
Comments 6: Line 385 - the References section is not made according to IJMS guidelines and should be corrected by the authors.
Response 6: Thank you for pointing this out. I have corrected the references section according to IJMS guidelines.
Round 2
Reviewer 2 Report
Comments and Suggestions for Authors
The paper is essentially the same, no changes have been made to clarify my main doubts. The answers were given only in the letter to the reviewer, without affecting the text.
In detail:
1) Although the authors claim that they identified and tested 17 PNC analogues, in the end the only one of interest was Phenazine-1-carboxamide, for which effects on R. solani and X. oryzae are already known. The title reads "Discovery of Phenazine-1- carbox-amide from Pseudomonas aeruginosa". What is the discovery? What did you find from the W-126 strain? This doesn't seem like an interesting discovery to me. From my point of view, there is no problem testing it on other pathogens as the authors did, but looking for it and finding it in a microorganism that we already know is capable of producing it, it is not interesting. Would the innovative element be just the fact that the microorganism in question was found in a medicinal plant? However, I don't see the usefulness for the purposes of the experimentation that was reported. If you had simply bought or extracted Phenazine-1-carboxamide from a culture of P. aeruginosa you could have performed the same test and obtained the same results.
2) the author answer to my concern about treatment comparison ("We just want to compare our compound with a commercial biocontrol agent") does not follow research practices in the field of plant pathology. In agricultural research, biocontrol agents are compared to each other, as well as molecules (BCA vs BCA, or chemical molecule vs chemical molecule). What can be done, in addition to a comparison between at least 2 molecules, is to add a third factor (eg. a biocontrol agent, with a set made by molecule 1 vs molecule 2 vs BCA), but the comparison cannot be limited to just two treatments as different in the way they act (as molecule vs BCA).
3) The explanation given on zhuorun ("Zhuorun is the commercial biocontrol agent of Bacillus amyloliquefaciens QST71") does not resolve the doubts I raised. The reader would not be able to repeat the experiment with the information reported in the text, and this is an obligation in the context of research. Why not give detailed references? I repeat that I was not able to find the product on the manufacturer's website. So the authors should at least include a link that allows the readers to view and purchase the product used.
The only element that seems to me to make the investigation into W-126 interesting is the test relating to the inhibitory capacity of the compounds found. But even in this case there are two things that don't convince me and that I discovered in this second step of the revision given the insistence of the authors:
1) Why in table 2 there are 19 PCN analogues while in supplementary S3 they are 17? The authors also write 17 in the response letter to my criticisms. Maybe because Streptomycin is a control, as reported at line 370? If so (and I am convinced that this is the case...), a compound is still missing and in any case the table header is wrong since it reports "PCN analogues" but they are not all PCN analogues.
2) The authors on line 168 of the revised document report "PCN showed a high inhibitory rate of all the PCN analogues against R. solanacearum (Table 2)" but this is not true. Inhibition by PCN is 68.18%, but Phenazine methosulfate reaches 88%. Is it not considered because the effect on M. incognita is limited? Nonetheless, the sentence is wrong. These are further signs that the article is poorly written in the M&Ms and results section.
Author Response
1) Although the authors claim that they identified and tested 17 PNC analogues, in the end the only one of interest was Phenazine-1-carboxamide, for which effects on R. solani and X. oryzae are already known. The title reads "Discovery of Phenazine-1-carboxamide from Pseudomonas aeruginosa". What is the discovery? What did you find from the W-126 strain? This doesn't seem like an interesting discovery to me. From my point of view, there is no problem testing it on other pathogens as the authors did, but looking for it and finding it in a microorganism that we already know is capable of producing it, it is not interesting. Would the innovative element be just the fact that the microorganism in question was found in a medicinal plant? However, I don't see the usefulness for the purposes of the experimentation that was reported. If you had simply bought or extracted Phenazine-1-carboxamide from a culture of P. aeruginosa you could have performed the same test and obtained the same results.
Thank you for pointing this out. To the first question, “What is the discovery? What did we find from the W-126 strain?” Although phenazine-1-carboxamide (PCN) has been identified and reported with effects on R. solani and X. oryzae, but the activity of PCN against R. solanacearum and M. incognita has not been reported. It is the first report to systematicly separate and use PCN to control of bacterial wilt and root-knot nematode diseases. For the second question, we did not simply buy or extract PCN from a culture of P. aeruginosa, but through systematic separation and extraction. We selected out the most active endophyte W-126 from 209 biocontrol bacteria through observation the anti-bacterial wilt zone in the beginning. Based on active compound tracking principles, the compound PCN was isolated from dozens components of endophyte W-126. Then the activity of 17 PNC analogues against R. solanacearum and M. incognita were tested. PNC showed high activity against both R. solanacearum and M. incognita of all the 17 PNC analogues. In addition, we also tested the antibacterial activity of the PCN against other 5 phytopathogenic bacteria. It was interesting that PCN showed the highest activity against R. solanacearum. These results showed that PCN has great activity difference and particularity in antagonizing various bacterial pathogens. So, we have further modified the introduction (line 84) and discussion part (line 231-235, line 247-248, line 251-252 and line 271-282) of the article to make the novelty and enrichment of the article more prominent.
2) the author answer to my concern about treatment comparison ("We just want to compare our compound with a commercial biocontrol agent") does not follow research practices in the field of plant pathology. In agricultural research, biocontrol agents are compared to each other, as well as molecules (BCA vs BCA, or chemical molecule vs chemical molecule). What can be done, in addition to a comparison between at least 2 molecules, is to add a third factor (eg. a biocontrol agent, with a set made by molecule 1 vs molecule 2 vs BCA), but the comparison cannot be limited to just two treatments as different in the way they act (as molecule vs BCA).
Thank you for pointing this out. I agree with this comment that biocontrol agents are compared to each other in addition to a comparison between at least 2 molecules, is to add a third factor. In fact, we did pot experiment to evaluate the control efficiency of PCN on controlling complex disease of R. solanacearum and M. incognita. We also used the fermentation broth of strain W-126 and W-33 as treatments, a concentration with 10 μg/mL of abamectin as positive control against M. incognita, and a concentration with 100 μg/mL of streptomycin as positive control against R. solanacea. In addition, the biocontrol agent Bacillus amyloliquefaciens QST713 has been registered as a microbial pesticide in China. We have added the treatments of W-126, abamectin and streptomycin in the section of 2.7 (line 181-187, line 192-196, and line 197-203), 4.8 (line 408-415, line 417-423, and line 427-428), Table 3 and Supplementary Figure S1.
3) The explanation given on zhuorun ("Zhuorun is the commercial biocontrol agent of Bacillus amyloliquefaciens QST71") does not resolve the doubts I raised. The reader would not be able to repeat the experiment with the information reported in the text, and this is an obligation in the context of research. Why not give detailed references? I repeat that I was not able to find the product on the manufacturer's website. So the authors should at least include a link that allows the readers to view and purchase the product used.
Thank you for pointing this out. I agree with comment. Zhuorun" is the commercial biocontrol agent of Bacillus amyloliquefaciens QST713. I have added a link that allows the readers to view and purchase the product (line 421-422). I want to use B. amyloliquefaciens QST713 as a positive control both against Ralstonia solanacearum and Meloidogyne incognita. Because B. amyloliquefaciens QST713 was reported with the ability against R. solanacearum and M. incognita. We also added detailed references (line 423).
The only element that seems to me to make the investigation into W-126 interesting is the test relating to the inhibitory capacity of the compounds found. But even in this case there are two things that don't convince me and that I discovered in this second step of the revision given the insistence of the authors:
1) Why in table 2 there are 19 PCN analogues while in supplementary S3 they are 17? The authors also write 17 in the response letter to my criticisms. Maybe because Streptomycin is a control, as reported at line 370? If so (and I am convinced that this is the case...), a compound is still missing and in any case the table header is wrong since it reports "PCN analogues" but they are not all PCN analogues.
Thank you for pointing this out. Yes, streptomycin is a control against R. solanacearum and abamectin is as positive control against M. incognita. This is not an accurate description and we have changed the table header (line 173).
2) The authors on line 168 of the revised document report "PCN showed a high inhibitory rate of all the PCN analogues against R. solanacearum (Table 2)" but this is not true. Inhibition by PCN is 68.18%, but Phenazine methosulfate reaches 88%. Is it not considered because the effect on M. incognita is limited? Nonetheless, the sentence is wrong. These are further signs that the article is poorly written in the M&Ms and results section.
Thank you for pointing this out. I agree with your comment. Although inhibition by phenazine methosulfate reaches 82.77%, but the corrected mortality rate of it on M. incognita is only 16.67%. The inhibition rate by PCN is 68.18% with the second highest inhibition rate of all the 17 PCN analogues. In addition, the corrected mortality rate of PCN on M. incognita is 54.62%, with the third highest corrected mortality of all the 17 PCN analogues. We also changed the description here (line 168-169).
Round 3
Reviewer 2 Report
Comments and Suggestions for Authors
In my opinion the Authors have not yet understood my doubts expressed in the first question. I am very clear about what was done, but I continue to consider the two studies (the one on the search for PCN from W-126 and the one on the Phenazine-1-carbox-amide test) to be unrelated, with the second being able to be conducted without the first part.
Nonetheless, this is not the most serious aspect, and I can accept the Authors' point of view.
What still doesn't convince me are the changes that have been made to the text. In itself, the change in Table 3 would answer my doubts perfectly. In my opinion the Authors have not yet understood my doubts expressed in the first question. I am very clear about what was done, but I continue to consider the two studies (the one on the search for PCN from W-126 and the one on the Phenazine-1-carbox-amide test) to be unrelated, with the second being able to be conducted without the first part.
Nonetheless, this is not the most serious aspect, and I can accept the authors' point of view.
What still doesn't convince me are the changes that have been made to the text. In itself, the change in Table 3 would answer my doubts perfectly. I would therefore have been inclined to accept the paper. The problem is that the experiment, according to the authors, requires at least a month (L. 426), not counting data analysis and writing. But the authors, since my last review, have only had about ten days. How was this possible? Did the authors already have the data available but had not included them in the first version of the paper because they thought they were not useful? This seems strange, given that the Authors, in their responses to my comments, claim to be aware of how the comparison should be made.
Finally I must be very unlucky, because the link reported in the article for the Zhuorun does not seem to work (the Bayer site reports "The document you requested could not be found on this server"). Could it be an access problem between international servers? This time too it seems strange to me, but in any case the reference must be accessible to every user in the world, not just from China.
Author Response
In my opinion the Authors have not yet understood my doubts expressed in the first question. I am very clear about what was done, but I continue to consider the two studies (the one on the search for PCN from W-126 and the one on the Phenazine-1-carbox-amide test) to be unrelated, with the second being able to be conducted without the first part.
Nonetheless, this is not the most serious aspect, and I can accept the Authors' point of view.
Thank you for pointing this out and thank you for believing our searching the active compound through systematic separation and extraction. And that's exactly what we did. We didn't want to study W-126 at first when we identified it as Pseudomonas aeruginosa. But we found that the three strains W-126, W-125, and W-54 which with best antagonistic activity against bacterial wilt were all P. aeruginosa. The antagonistic activity against bacterial wilt of W-126 extracts from various organic solvents also indicated there are still small molecule active compounds which have not been evaluated.
What still doesn't convince me are the changes that have been made to the text. In itself, the change in Table 3 would answer my doubts perfectly. In my opinion the Authors have not yet understood my doubts expressed in the first question. I am very clear about what was done, but I continue to consider the two studies (the one on the search for PCN from W-126 and the one on the Phenazine-1-carbox-amide test) to be unrelated, with the second being able to be conducted without the first part.
Nonetheless, this is not the most serious aspect, and I can accept the authors' point of view.
Thank you for your approval of my revision work. We really found the antagonistic effects of W-126 first and then followed the active compound tracing process. And thank you for believing our searching the active compound through systematic separation and extraction.
What still doesn't convince me are the changes that have been made to the text. In itself, the change in Table 3 would answer my doubts perfectly. I would therefore have been inclined to accept the paper. The problem is that the experiment, according to the authors, requires at least a month (L. 426), not counting data analysis and writing. But the authors, since my last review, have only had about ten days. How was this possible? Did the authors already have the data available but had not included them in the first version of the paper because they thought they were not useful? This seems strange, given that the Authors, in their responses to my comments, claim to be aware of how the comparison should be made.
Thank you for pointing this out. And thank you for pointing the research practices in the field of plant pathology in the second review comments. I ignored the usefulness and importance of the abamectin and streptomycin sulfate treatments in the first version of the paper. In fact, we did many treatments to evaluate the control efficiency of PCN on controlling complex disease of R. solanacearum and M. incognita. We also used the fermentation broth of strain W-126 and W-33 as treatments, used abamectin, streptomycin and Bacillus QST713 as positive control. We also evaluated the effect of different number of nematodes inoculation on the incidence degree of bacterial wilt. I didn't carefully think the pot results at the beginning and some of the data wasn't put in.
Finally I must be very unlucky, because the link reported in the article for the Zhuorun does not seem to work (the Bayer site reports "The document you requested could not be found on this server"). Could it be an access problem between international servers? This time too it seems strange to me, but in any case the reference must be accessible to every user in the world, not just from China.
Thank you for pointing this out. The ”Zhuorun" is the commercial biocontrol agent of Bacillus amyloliquefaciens QST713 and it has been registered as a microbial pesticide in China. I am sorry for you could not find it on the server website because of access problem between international servers. Bacillus QST713 is a strain that has been commercialized worldwide by Bayer. I think the products of this strain are certainly available in many different countries or regions.